# Exosomes: Key Factors in Ovarian Cancer Peritoneal Metastasis and Drug Resistance

**DOI:** 10.3390/biom14091099

**Published:** 2024-09-02

**Authors:** Ming Shao, Yunran Gao, Xiling Xu, David Wai Chan, Juan Du

**Affiliations:** 1Guangdong Key Laboratory for Biomedical Measurements and Ultrasound Imaging, National-Regional Key Technology Engineering Laboratory for Medical Ultrasound, School of Biomedical Engineering, Shenzhen University Medical School, Shenzhen 518060, China; shaoming007@163.com; 2Department of Central Laboratory, The Second Affiliated Hospital, School of Medicine, The Chinese University of Hong Kong, Shenzhen & Longgang District People’s Hospital of Shenzhen, Shenzhen 518172, China; 3School of Basic Medical Sciences, Anhui Medical University, 81 Meishan Road, Hefei 230032, China; yunrangao0523@gmail.com (Y.G.); 2245010210@stu.ahmu.edu.cn (X.X.); 4School of Medicine, The Chinese University of Hong Kong, Shenzhen 518172, China

**Keywords:** ovarian cancer, exosomes, peritoneal metastasis, drug resistance, epithelial–mesenchymal transition

## Abstract

Ovarian cancer remains a leading cause of death among gynecological cancers, largely due to its propensity for peritoneal metastasis and the development of drug resistance. This review concentrates on the molecular underpinnings of these two critical challenges. We delve into the role of exosomes, the nano-sized vesicles integral to cellular communication, in orchestrating the complex interactions within the tumor microenvironment that facilitate metastatic spread and thwart therapeutic efforts. Specifically, we explore how exosomes drive peritoneal metastasis by promoting epithelial–mesenchymal transition in peritoneal mesothelial cells, altering the extracellular matrix, and supporting angiogenesis, which collectively enable the dissemination of cancer cells across the peritoneal cavity. Furthermore, we dissect the mechanisms by which exosomes contribute to the emergence of drug resistance, including the sequestration and expulsion of chemotherapeutic agents, the horizontal transfer of drug resistance genes, and the modulation of critical DNA repair and apoptotic pathways. By shedding light on these exosome-mediated processes, we underscore the potential of exosomal pathways as novel therapeutic targets, offering hope for more effective interventions against ovarian cancer’s relentless progression.

## 1. Introduction

Ovarian cancer continues to represent one of the most lethal gynecological malignancies, largely due to its asymptomatic progression and advanced stage at diagnosis [1,2]. The effective treatment of this disease is impeded by two significant obstacles: metastasis and drug resistance [3,4,5]. These factors often result in poor prognoses and high mortality rates [6]. A comprehensive understanding of the underlying mechanisms that drive these processes is essential for the development of novel therapeutic strategies. Recent research has underscored the pivotal role of exosomes, nano-sized vesicles secreted by cells, in orchestrating diverse aspects of ovarian cancer progression, including metastasis and drug resistance [7,8].

Exosomes, which typically range in diameter from 30 to 150 nm, are encapsulated by a lipid bilayer membrane, and carry a diverse array of biologically active molecules, including deoxyribonucleic acid (DNA), various forms of ribonucleic acid (RNA), proteins, and cytokines [9]. These vesicles are not mere cellular waste; rather, they play a pivotal role in intercellular communication, influencing the behavior and fate of recipient cells [10,11].

In the context of cancer, exosomes have been demonstrated to facilitate tumor growth, metastasis, and the development of drug resistance by transferring oncogenic molecules to target cells [12]. This intercellular communication is of particular relevance in the context of the tumor microenvironment, where exosomes have the capacity to modulate immune responses, promote angiogenesis, and alter the behavior of stromal cells, thereby creating a supportive niche for tumor cells [13,14].

The majority of ovarian cancer metastases occur in the peritoneal cavity, with cancer cells spreading throughout the abdominal region [15]. Exosomes derived from ovarian cancer cells have been demonstrated to play a role in this process through the induction of epithelial–mesenchymal transition (EMT) in peritoneal mesothelial cells, thereby enhancing their invasive capabilities [16,17]. Additionally, exosomes contribute to the remodeling of the extracellular matrix and the creation of a pre-metastatic niche, thereby further facilitating cancer cell dissemination [3]. Additionally, these vesicles play a pivotal role in angiogenesis, a process vital for supplying nutrients and oxygen to growing tumors, and in immune evasion, enabling cancer cells to evade detection and destruction by the host’s immune system [18,19].

Another significant challenge in the treatment of ovarian cancer is drug resistance, which exosomes are known to facilitate. They are capable of sequestering and expelling chemotherapeutic drugs from tumor cells, thereby reducing the intracellular concentration of these agents and diminishing their efficacy. Additionally, exosomes transfer drug resistance-associated molecules, such as microRNAs and proteins, to drug-sensitive cells, thereby disseminating resistance traits across the tumor population. Furthermore, exosomes regulate DNA damage repair pathways and apoptotic responses, thereby promoting the survival and proliferation of drug-resistant cancer cells [20,21,22].

This review endeavors to dissect the sophisticated molecular mechanisms by which exosomes contribute to peritoneal metastasis and drug resistance in ovarian cancer. By delving into the cellular and molecular dialogues facilitated by exosomes within the tumor microenvironment, we aim to unravel how these vesicles mediate the transformation of normal peritoneal cells, foster the degradation and remodeling of the extracellular matrix, and establish new vascular networks, all of which are essential for metastatic colonization. Simultaneously, we aim to clarify the molecular stratagems employed by exosomes to circumvent chemotherapeutic interventions, including the direct expulsion of drugs, the transfer of resistance-conferring genetic material, and the alteration of cell survival pathways. Through a detailed understanding of these processes, this review aspires to highlight novel molecular targets within the exosome-mediated pathways, paving the way for innovative therapeutic strategies that could significantly ameliorate the prognosis and treatment of ovarian cancer.

## 2. Biological Characterization of Exosomes

Exosomes are nano-sized vesicles, typically 30 to 150 nm in diameter, encapsulated by a lipid bilayer membrane (Figure 1). The process of exosome production is primarily divided into four stages: invagination of the cell membrane, formation of endosomes, development of multivesicular bodies (MVBs), and ultimately, fusion of these bodies with the plasma membrane followed by release into the cytosol as exosomes [23]. During exosome formation, the plasma membrane invaginates in a cup-like structure, wrapping cell surface proteins and soluble proteins associated with the extracellular environment to form an early sorting endosome (ESE), and the Golgi apparatus and endoplasmic reticulum assist in ESE formation. Later, the ESE matures into late sorting endosomes (LSE), and LSE invagination leads to the formation of intracellular multivesicular bodies (MVBs) containing intraluminal vesicles (ILVs). Then, MVBs fuse with the cell membrane and cytosolizes to secrete exosomes [9]. These exosomes are rich in diverse biologically active molecules, including DNA, various forms of RNA, proteins, and cytokines [9,24,25]. The exosome membranes are characterized by specific marker proteins such as CD9, CD63, and CD81, which facilitate their recognition and binding to target cells [26,27,28].

Recent studies have emphasized the pivotal role of exosomes in the tumor microenvironment, where they influence critical aspects such as tumor cell proliferation, migration, invasion, angiogenesis, immune modulation, and metabolic reprogramming [25]. These processes collectively contribute to drug resistance and metastasis. Over the past decade, exosome research has made considerable strides, highlighting their promise as novel diagnostic biomarkers and therapeutic vehicles [29,30,31].

## 3. Mechanisms of Exosomes in Ovarian Cancer Metastasis

Ovarian cancer metastasis and drug resistance represent significant challenges in the effective treatment of this malignancy. Exosomes, which play a critical role in these processes, exert substantial influence on ovarian cancer progression and therapeutic outcomes [21,32,33]. These nano-sized vesicles deliver bioactive molecules, such as proteins, RNAs, lipids and other factors, to recipient cells through membrane fusion, endocytosis, and receptor-mediated uptake. This intercellular communication promotes tumor cell proliferation, migration, invasion, and metastasis [34,35].

Exosomes can also affect other cells in the tumor microenvironment and promote tumor progression by altering the original biological phenotypes of these receptor cells; for example, tumor cell-derived exosomes promote endothelial cell proliferation and neovascularization [36]. Exosomes of mesenchymal cell origin also promote tumor progression by altering tumor cell phenotypes; for example, exosomes secreted by cancer-associated fibroblasts (CAFs) promote tumor metastasis and chemoresistance by enhancing tumor cell stemness and epithelial–mesenchymal transition (EMT) [20]. Thus, exosomes are critical regulators of tumor microenvironment modulation and cancer metastasis (Figure 2 and Table 1).

### 3.1. Exosomes Regulate Cancer-Associated Fibroblasts (CAFs) to Promote Ovarian Cancer Metastasis

Cancer-associated fibroblasts (CAFs) represent a critical component of the tumor microenvironment, exerting a substantial influence on tumor progression [52]. CAFs facilitate tumor growth through the remodeling of the extracellular matrix (ECM); the secretion of growth factors, cytokines, and chemokines; the regulation of tumor metabolism; and the induction of angiogenesis. They also modulate immune cell function. They alter the tumor immune microenvironment to suppress anti-tumor responses [53].

Tumor cells can influence the function of CAFs through exosomes, transforming them into “helpers” that facilitate tumor growth, drug resistance, and metastasis [54]. Exosomes from ovarian cancer cells have been shown to activate fibroblasts into a CAF-like state, enhancing their proliferation, motility, invasiveness, and enzyme expression [55]. Specifically, hsa-miR-141-3p (miR-141) secreted by ovarian cancer cells is transferred to stromal fibroblasts via exosomes. By targeting YAP1, a key effector of the Hippo signaling pathway, miR-141 downregulates YAP1 expression and increases the secretion of the pro-inflammatory chemokine GROα in stromal fibroblasts. This reprogramming of stromal fibroblasts into pro-inflammatory CAFs facilitates the formation of a metastatic niche, thus promoting ovarian cancer metastasis [3].

On the other hand, CAF-derived exosomes may also be involved in cancer cell proliferation, metastasis, and drug resistance. Growing evidence suggests that exosomes mediate the interaction between CAFs and other cells in the tumor microenvironment, contributing to tumor metastasis [56]. In ovarian cancer studies, the downregulation of miR-29c-3p in exosomes from CAFs has been associated with the promotion of peritoneal metastasis. CAF-derived exosomes with low miR-29c-3p levels enhance MMP2 expression in ovarian cancer cells, accelerating intraperitoneal metastasis [43]. Additionally, high levels of miR-21 in CAF-derived exosomes from the omental tumor microenvironment are linked to ovarian cancer metastasis. These exosomes transfer miR-21 to ovarian cancer cells, directly binding to APAF1 and downregulating its expression, which inhibits apoptosis and confers paclitaxel resistance, thus promoting distant metastasis [44]. It was found that the combination of the chemotherapeutic drug gemcitabine and the exosome inhibitor GW4869 significantly reduced the number of exosomes released by CAFs, resulting in a better therapeutic effect [57]. To summarize, exosomes provide an important link between ovarian cancer cells and CAFs. Exosomes from ovarian cancer cells convert CAFs into metastasis-supporting factors, while exosomes secreted by CAFs further enhance tumor resistance and metastasis.

### 3.2. Exosomes Promote Ovarian Cancer Metastasis by Inducing Epithelial–Mesenchymal Transition (EMT) in Peritoneal Mesothelial Cells

Ovarian cancer metastasis is unique compared to other cancers due to its primary mode of spread, which is peritoneal metastasis. The peritoneum, mainly composed of mesothelial cells and a small amount of connective tissue, is a crucial barrier preventing ovarian cancer cells from invading the retroperitoneal cavity and protecting underlying tissues [58]. However, within the tumor microenvironment, ovarian cancer cells secrete oncogenic factors that induce a mesothelial-to-mesenchymal transition (MMT) in normal mesothelial cells, transforming them into cancer-associated mesothelial cells (CAMs) [59]. This transformation results in the loss of the protective barrier function of mesothelial cells. CAMs exhibit epithelial–mesenchymal transition (EMT) features, such as increased expression of fibronectin, α-SMA, vimentin, and decreased E-cadherin levels. These CAMs further secrete cytokines, chemokines, and exosomes, enhancing ovarian cancer cells’ metastatic potential [60,61,62].

Recent studies have highlighted the significant role of exosomes in mediating communication between ovarian cancer cells and CAMs, thereby facilitating ovarian cancer metastasis. For instance, Koji Nakamura et al. discovered that exosomes derived from ovarian cancer cells were enriched with CD44. When co-cultured with human peritoneal mesothelial cells (HPMCs), these exosomes induced high levels of CD44 expression in HPMCs, leading to morphological changes and the adoption of an EMT phenotype in these cells. This elevated CD44 expression promoted peritoneal metastasis by inducing HPMCs to secrete MMP9, thereby disrupting the peritoneal mesothelial barrier and enhancing the invasive capacity of cancer cells [37].

Further research by Lingling Gao et al. demonstrated that annexin A2 (ANXA2), a membrane-associated protein in ovarian cancer cells, could be transferred to human peritoneal mesothelial cells (HMRSV5) via exosomes. ANXA2 regulated morphological changes and fibrosis in these cells, promoting EMT and extracellular matrix degradation through the PI3K/AKT/mTOR pathway. This process remodels the microenvironment, creating favorable conditions for ovarian cancer metastasis [38].

Moreover, Xiaoduan Li et al. identified the ITGA5B1/AEP complex as highly expressed in the exosomes found in the serum and ascites of ovarian cancer patients. This complex can be transferred to mesothelial cells via exosomes, activating the FAK/Akt/Erk pathway, thereby promoting EMT in mesothelial cells. This activation regulates mesothelial cell proliferation and migration, ultimately influencing peritoneal metastasis of ovarian cancer [39]. Intriguingly, a recent study reported a novel mechanism where exosomes carrying MMP1 mRNA in the ascites of ovarian cancer patients induce apoptosis in mesothelial cells, leading to the destruction of the peritoneal barrier, further facilitating peritoneal metastasis [63].

Exosomes play a key role in communication between ovarian cancer cells and mesothelial cells, promote EMT in mesothelial cells, and greatly facilitate peritoneal metastasis of ovarian cancer by secreting MMP proteins and other factors that disrupt the peritoneal barrier. Understanding these mechanisms is critical for designing cutting-edge therapies to address ovarian cancer spread and improve patient survival. However, we should not only focus on exosomes that promote EMT. An increasing body of research now indicates that exosomes also play a role in inhibiting epithelial–mesenchymal transition (EMT) [64]. For example, it has been found that LBH-enriched exosomes inhibit tumor EMT by down-regulating VEGFA signaling [65]. In addition, it has been found that exosomes derived from mesenchymal stem cells (MSCs) have a role in inhibiting EMT in tumor cells, which offers a new strategy for cancer treatment [66]. These studies suggest that exosomes may function as a double-edged sword in regulating peritoneal metastasis of ovarian cancer through epithelial–mesenchymal transition (EMT). Specifically, exosomes may promote the EMT process of ovarian cancer cells under certain conditions, thereby enhancing their invasiveness and metastatic potential, leading to disease progression. Conversely, under other conditions, exosomes might inhibit the EMT process, reducing the invasiveness and metastasis of cancer cells, thus alleviating the disease to some extent. Therefore, the role of exosomes in ovarian cancer peritoneal metastasis is complex and variable, depending on their microenvironment and specific regulatory mechanisms. These findings suggest that when considering exosomes as therapeutic targets, their dual role must be carefully considered to avoid potential adverse effects.

### 3.3. Exosome-Induced Angiogenesis Promotes Ovarian Cancer Metastasis

Angiogenesis is a critical factor in the progression of tumor metastasis and the development of drug resistance [67]. Blood vessels, formed from a single layer of endothelial cells, provide essential nutrients to tumor cells, creating a favorable environment for their growth and aiding in immune evasion [12]. However, the structure and function of tumor-induced neovascularization are often abnormal, characterized by an incomplete stroma, and a propensity for leakage. This allows tumor cells to penetrate the vascular endothelium, enter the bloodstream, and colonize other body parts, thus facilitating metastasis [68,69]. Increasing evidence underscores the significant role of vascular endothelial cells in promoting tumor cell metastasis.

Tumor-associated exosomes are known to carry various proteins, RNAs, and factors such as vascular endothelial growth factor (VEGF), fibroblast growth factor (FGF), transforming growth factor β (TGF-β), and tumor necrosis factor (TNF). These components stimulate angiogenesis and alter the permeability of vascular endothelial cells, thereby promoting the establishment of pre-metastatic niches that are favorable to tumor cells [70,71]. For instance, ovarian cancer cell-derived exosomes carrying the PKR1 protein have been shown to activate the PKR1 signaling pathway by inducing STAT3 phosphorylation in human umbilical vein endothelial cells (HUVECs). This activation promotes HUVEC migration and tube formation, elucidating the mechanism by which ovarian cancer cell-derived exosomes facilitate tumor angiogenesis [40].

Further studies have identified that miR-205 is highly expressed in the tumor tissues and sera of ovarian cancer patients, and its elevated levels in circulating exosomes are associated with tumor metastasis [41]. Mechanistically, exosome-dependent secretion of miR-205 from ovarian cancer cells to neighboring vascular endothelial cells, mediated by endocytosis via lipid rafts, regulates the PTEN-AKT pathway, inducing angiogenesis and subsequent tumor metastasis [41]. Additionally, Tang, M.K.S., et al. discovered that soluble E-cadherin (sE-cad), abundantly present in the malignant ascites of ovarian cancer patients, is a potent inducer of angiogenesis. These sE-cad-positive exosomes form heterodimers with VE-cadherin on endothelial cells, triggering a novel sequential activation of β-catenin and NF-κB signaling pathways, thereby enhancing angiogenesis in ovarian cancer [42].

In summary, exosomes exert a profound influence on the angiogenic and metastatic processes in ovarian cancer by transporting essential pro-angiogenic factors and regulatory molecules. The insights gained from studying the exosome-mediated signaling pathways and their effects on vascular endothelial cells offer a valuable understanding of the progression of ovarian cancer and present potential avenues for developing novel therapeutic strategies. Exosomes derived from miR-16-containing mesenchymal stem cells have been reported to down-regulate vascular endothelial growth factor expression, thereby inhibiting angiogenesis in vitro and in vivo [72]. In addition, the combination of exosomes with monoclonal antibodies against vascular endothelial growth factor provides a new idea for cancer treatment [73].

### 3.4. Exosomes Regulate Immune Cells to Promote Ovarian Cancer Metastasis

Immune cells, including lymphocytes, dendritic cells, monocytes, macrophages, granulocytes, and mast cells, play vital roles in antigen recognition, presentation, and immune response [74]. While these cells typically act as defenders against cancer, their function can be altered within the tumor microenvironment, aiding immune escape and promoting drug resistance and metastasis [75,76].

Exosomes mediate intercellular communication and significantly influence the tumor immune microenvironment [77]. They are involved in both immune stimulation and immunosuppression within ovarian cancer. On the one hand, exosomes can stimulate immune cells to exert anti-tumor functions by secreting biological factors and transmitting bioactive substances. On the other hand, exosomes secreted by ovarian cancer cells can shift immune cells from an activated state to an immunosuppressive one, facilitating tumor growth, drug resistance, and metastasis by evading immune surveillance [78,79]. Exosomes carry immunoreactive molecules, including major histocompatibility complex (MHC I), heat shock proteins (HSPs), and CD81, which stimulate anti-tumor immune responses [80]. However, they can also enhance immune escape by promoting immunosuppression, hereby promoting tumor development and metastasis [81].

CD4(+) and CD8(+) T lymphocytes are crucial for specific anti-tumor immunity. Emerging evidence suggests that exosomes can carry various immunosuppressive signals to inhibit T-cell proliferation and function, contributing to tumor immunity [82]. Exosomes isolated from the ascites of ovarian cancer patients can inhibit T-cell receptor (TCR)-dependent T-cell activation, affecting endpoints such as NF-κB and NFAT translocation, CD69 and CD107a upregulation, cytokine production, and cellular proliferation. This immune suppression is reversible, and blocking these exosomes can reactivate anti-tumor responses in suppressed tumor-associated T-cells [45].

The programmed death ligand-1/programmed death receptor-1 (PD-L1/PD-1) signaling pathway is crucial to tumor immunosuppression. It inhibits T lymphocyte activation and enhances tumor immune tolerance, facilitating immune escape [83,84,85]. For example, elevated reactive oxygen species (ROS) in ovarian cancer downregulate exosomal miR-155-5p, increasing PD-L1 expression and reducing CD8(+) T lymphocytes, thereby aiding immune escape [46]. Cisplatin-resistant ovarian cancer cells secrete exosomes carrying plasma gelsolin (pGSN), which induce CD8(+) T lymphocyte apoptosis, decrease IFN-γ secretion, and increase the glutathione (GSH) content in ovarian cancer cells, enhancing immune resistance [86].

Studies on the impact of exosomes on the tumor immune microenvironment in ovarian cancer have mainly focused on tumor-associated macrophages (TAMs) [87]. TAMs release exosomes, cytokines, chemokines, and enzymes that enhance ovarian cancer cell invasiveness and chemoresistance, playing a crucial role in peritoneal metastasis by aiding tumor cell adhesion and sphere formation [88]. M2-type macrophages, predominantly found among TAMs, are pro-carcinogenic [89]. Hypoxia increases the secretion of exosomes carrying miR-21-3p, miR-125b-5p, and miR-181d-5p from ovarian cancer cells, promoting M2 macrophage polarization and furthering tumor cell proliferation and migration [47]. Elevated ROS levels in ovarian cancer cells reduce exosomal miR-155-5p uptake by TAMs, upregulating immunosuppressive factors like PD-L1 and promoting immune escape [46]. Additionally, ovarian cancer cells with high ETS proto-oncogene 1 (ETS1) expression release laminin-rich exosomes that promote M2 macrophage polarization and CXCL5 and CCL2 secretion via the integrin αvβ5/akt/sp1 pathway, supporting metastasis [48].

TAM-derived exosomal miRNAs also play significant roles in ovarian cancer metastasis. For instance, miR-221-3p in M2-TAM exosomes inhibits CDKN1B, promoting tumor cell proliferation and peritoneal metastasis [49]. miR-29a-3p in TAM exosomes enhances PD-L1 expression in ovarian cancer cells via the FOXO3-AKT/GSK3β pathway, aiding immune escape [50]. TAMs also interact with other tumor microenvironment cells, as seen in exosome-mediated miR-29a-3p and miR-21-5p interactions with T-cells, leading to STAT3 inhibition and Treg/Th17 ratio imbalance, fostering an immunosuppressive environment and promoting metastasis [51].

These findings provide new insights into the role of immune cells in the ovarian cancer tumor microenvironment, highlighting potential diagnostic markers and therapeutic targets. However, in ovarian cancer research, we have found that most studies have focused primarily on exosomes inhibiting immune responses, while there is a lack of attention to the other side of the coin, i.e., that exosomes can also activate immunity [90]. For example, it was found that exosomes are involved in antigen presentation to modulate the immune response. In a pancreatic cancer study, tumor-derived exosomes interacted with antigen-presenting cells (APCs) to efficiently kill tumors by activating tumor antigen-specific cytotoxic T-cell (CTL) responses [91]. Thus, further comprehensive investigation of the mechanisms of exosomal influence on the immune system is warranted.

## 4. Mechanisms of Exosomes in Ovarian Cancer Drug Resistance

Exosome secretion is a universal capability of all cell types, enabling them to mediate intercellular communication and modulate the function of recipient cells. These functions are inextricably linked to many physiological processes and the progression of various diseases [92,93,94,95]. For example, exosomes secreted by drug-resistant tumor cells can cause drug-resistant phenotypic changes in sensitive cells, and information transfer between tumor cells via exosomes allows tumor cells to escape immune killing [94] better.

They modulate immune cell activity and can alter the expression of drug-resistant genes and survival pathways, thereby affecting the sensitivity of tumor cells to chemotherapy and promoting the development of drug resistance [96,97,98]. Understanding the diverse mechanisms through which exosomes contribute to ovarian cancer drug resistance is crucial for developing novel therapeutic strategies to overcome these challenges (Figure 3 and Table 2).

### 4.1. Exosome-Promoted Drug Efflux Leads to Drug Resistance in Ovarian Cancer

Decreased intracellular concentrations of anti-cancer drugs contribute significantly to tumor cell drug resistance. Hydrophobic drugs and their breakdown products interact with exosome lipid membranes, becoming encapsulated and expelled from tumor cells. Understanding the mechanism of exosome-mediated drug exocytosis is crucial for developing therapies for chemotherapy-resistant ovarian cancer [12]. It is well documented that drug-resistant ovarian cancer cells secrete more exosomes than drug-sensitive cells, a phenomenon linked to the transmembrane protein TMEM205 [99,100]. Safaei et al. found that the cisplatin (CDDP)-resistant ovarian cancer cell line 2008/C13×5.25 released 2.6 times more CDDP in exosomes than CDDP-sensitive cells, with CDDP concentrated in lysosomes, indicating that exosomes contribute to resistance through abnormal protein sorting and release, leading to increased CDDP expulsion [101]. Under hypoxic conditions, ovarian cancer cells upregulate Rab27a and downregulate Rab7, LAMP1/2, and NEU-1, promoting exosome release. Hypoxic ovarian cancer cell exosomes carrying STAT3 and FAS significantly increase chemotherapy resistance in vitro [102].

Increased drug efflux is one of the causes of drug resistance in tumors. Increased exosome secretion leads to drug efflux, preventing chemotherapeutic agents from adequately killing tumor cells [110]. Combining the exosome release inhibitor GW4869 with chemotherapeutic drugs has been reported to restore chemosensitivity in tumors [111].

### 4.2. Exosomes Promote Drug Resistance in Ovarian Cancer by Modulating DNA Damage Repair Systems and Death Pathways

Exosomes can significantly contribute to the development of drug resistance by delivering drug resistance-associated molecules to drug-sensitive cells. Wang et al. reported that the expression of PANDAR (promoter of CDKN1A antisense DNA damage-activated RNA) is higher in cisplatin-resistant ovarian cancer tissues and cells from patients with wild-type p53 ovarian cancer than in cisplatin-sensitive cases [112]. Their studies showed that exosomes from ovarian cancer cell lines carrying PANDAR increased the SIRT4/SIRT6 mRNA proportion in ovarian cancer cells by interacting with the target gene SRSF9, significantly enhancing tumor cell survival and chemotherapy resistance in vitro [113].

DNA methyltransferase 1 (DNMT1) plays an essential role in maintaining genome-wide methylation during DNA replication and damage repair, and its oncogenic potential has been well documented [103]. Recently, DNMT1 was highly enriched in exosomes secreted by ovarian cancer cells. In vivo experiments demonstrated that DNMT1-containing exosomes promoted tumor resistance, while treatment with the exosome inhibitor GW4869 almost wholly restored the sensitivity of resistant cells [114]. Additionally, higher plasma gelsolin (pGSN) expression was observed in chemotherapy-resistant ovarian cancer cells compared to sensitive ones. This was mediated through exosomal secretion, which upregulated HIF1α expression and conferred cisplatin resistance to other chemotherapy-sensitive cells [115]. pGSN in exosomes also induced CD8(+) T-cell apoptosis, leading to reduced IFNγ secretion and increased glutathione (GSH) production in ovarian cancer cells, enhancing resistance to CDDP-induced cell death [86].

The role of tumor cell-derived exosomal miRNAs (exo-miRNAs) in fostering drug resistance has garnered significant attention. Exo-miRNAs regulate gene expression in target cells locally and systematically, influencing disease progression by modulating DNA damage repair systems and death pathways and promoting chemoresistance [116]. For example, miR-429 was highly expressed in multidrug-resistant SKOV3 cells and their secreted exosomes, compared to sensitive A2780 cells. miR-429 promoted A2780 cell proliferation and drug resistance by targeting the calcium-sensing receptor (CASR)/STAT3 pathway [104].

Interestingly, increased expression of circular RNA hsa_circ_0010467 was observed in exosomes from platinum-resistant ovarian cancer cells. RNA-binding protein AUF1 promotes the biogenesis of hsa_circ_0010467 in ovarian cancer, which activates the LIF/STAT3 signaling pathway by mediating the inhibitory effect of miR-637 on leukemia inhibitory factor (LIF), thereby promoting platinum resistance [105]. Exo-miRNAs also play crucial roles in inhibiting apoptosis and regulating DNA damage repair, contributing to drug resistance. For instance, Zou et al. reported that exosomal miR-6836 can be transferred to cisplatin-sensitive epithelial ovarian cancer (EOC) cells, promoting drug resistance by targeting DLG2 and enhancing Yap1 nuclear translocation, forming a TEAD1-regulated positive feedback loop that increases cell stemness and inhibits apoptosis [22].

Levels of miR-21 are markedly elevated in exosomes derived from cancer-associated adipocytes (CAA) and fibroblasts (CAF) harvested from the omental tumor milieu. These exosomes convey miR-21 to ovarian cancer cells, where it binds directly to APAF1, suppresses its expression, impedes apoptosis, and bestows resistance to paclitaxel [44]. Guo et al. found that cyclin-dependent kinase inhibitor 1A (CDKN1A) is highly expressed in cisplatin-sensitive ovarian cancer cells. CAF-derived exosomes carrying miR-98-5p increase OC cell proliferation and cell cycle entry, inhibit apoptosis, and promote cisplatin resistance by downregulating CDKN1A [106].

Moreover, exosomes originating from ovarian cancer cells are notably enriched with miR-21-3p, miR-21-5p, and miR-891-5p [117]. Quantitative MS/MS analyses have shown that miR-21-5p stimulates glycolysis and elevates the levels of ATP-binding cassette family proteins as well as detoxifying enzymes. Both miR-21-3p and miR-891-5p play roles in regulating proteins linked to DNA repair processes; collectively aiding in the development of carboplatin resistance in ovarian cancer [107]

In summary, exosomes contribute to drug resistance in ovarian cancer through complex interactions with DNA damage repair systems and apoptotic pathways. A deeper understanding of these mechanisms could help create innovative therapeutic approaches to circumvent chemoresistance in ovarian cancer. Interestingly, it was previously reported that miR-21, which is highly expressed in ovarian cancer, impedes tumor cell apoptosis and leads to paclitaxel resistance [44]. Then, we found that in another report, the researchers’ engineered exosome-based co-delivery system of the chemotherapeutic drug 5-FU and the miR-21 inhibitor oligodeoxynucleotide (miR-21i) efficiently promoted 5-FU-resistant HCT116 cells to undergo apoptosis and reverse drug resistance [118]. This makes us wonder whether an engineered exosome co-delivery system constructed in the same way as paclitaxel and miR-21i would be effective in reversing drug resistance in ovarian cancer.

### 4.3. Exosomes Regulate Immune Cells to Promote Drug Resistance in Ovarian Cancer

Recent studies investigating the interplay between tumor-associated immune cells and exosomes in ovarian cancer, particularly concerning mechanisms of drug resistance, are primarily centered on tumor-associated macrophages (TAMs) [119]. TAMs are the predominant immune cell population in the ovarian tumor microenvironment, characterized by high plasticity. They can be easily polarized into an immunosuppressive M2-like phenotype by colony-stimulating factor-1 released by tumor cells, which is closely linked to ovarian cancer progression and chemoresistance [87].

Previous studies have shown that ovarian cancer cell-derived exosomal miR-1246 confers chemoresistance to OC cells by directly targeting Cav1; when OC cells were co-cultured with macrophages, they transferred their oncogenic miR-1246 to M2-type TAMs, rather than M0-type TAMs. This suggests that ovarian cancer cells promote tumor resistance through exosomes transferred to adjacent infiltrating immune cells [108]. Conversely, exosomes isolated from OC cells treated with naphthoquinone shikonin (SK), which has anti-tumor effects, were found to inhibit M2 polarization of macrophages by blocking β-catenin activation mediated by exosomal galectin-2 (LGALS2). This reduced the infiltration of M2 macrophages in tumor tissues, providing a novel approach for immunotherapy against ovarian cancer resistance [120].

Furthermore, recent research indicates that hypoxic environments within epithelial ovarian cancer (EOC) cells initiate the recruitment of macrophages and stimulate their transformation into a phenotype resembling tumor-associated macrophages (TAMs). [121]. MiR-223 was found in high concentrations in TAM-derived exosomes produced under hypoxic conditions and could enhance drug resistance in epithelial ovarian cancer (EOC) cells through the PTEN-PI3K/AKT signaling pathway [109].

Despite these insights, there remains a significant gap in understanding the drug resistance mechanisms of immune cell-derived exosomes in the ovarian cancer tumor microenvironment beyond TAMs. In exosome-mediated drug resistance, research on other immune cells, such as lymphocytes, NK cells, dendritic cells, monocytes, granulocytes, and mast cells, is sparse and warrants further exploration.

## 5. The Potential of Exosomes in the Treatment of Ovarian Cancer

The significant role of exosomes within the ovarian cancer tumor microenvironment establishes them as effective therapeutic targets for therapy and treatment approaches [122,123]. The findings discussed earlier about exosomes promoting metastasis and drug resistance suggest potential therapeutic applications targeting exosomes in ovarian cancer. With their ability to target specific cells and deliver bioactive molecules, exosomes have become a focal point in drug delivery research owing to their excellent biocompatibility, high permeability, and low immunogenicity [9]. For instance, the cisplatin-loaded exosomes from umbilical cord blood-derived M1 macrophages could target ovarian cancer and reverse cisplatin resistance in vivo [124]. Exosomes can also serve as carriers of siRNA for ovarian cancer treatment [125]. Recently, cell-free immunotherapy has emerged as a noteworthy approach to treating ovarian cancer. In this domain, exosomes derived from immune cells are recognized for their capacity to modulate immune responses. Heyong Luo et al. have shown that using exosomes from NK cells to deliver cisplatin can rejuvenate the immune activity of NK cells in the tumor microenvironment, offering an innovative treatment method [126].

Cutting-edge technologies, like the M-Trap, have been designed to isolate T-cells, aiming to treat peritoneal metastases stemming from ovarian cancer [127]. Simone Pisano et al. pioneered the development of immune cell-derived exosome mimetics (IDEMs), introducing a new strategy to combat ovarian cancer [128]. Furthermore, Longxia Li et al. demonstrated that combined delivery of TP and miR-497 using exosome–liposome composite nanoparticles effectively overcome drug resistance in ovarian cancer [129].

Given that most ovarian cancer patients are diagnosed at advanced stages with extensive peritoneal metastases, the importance of targeted delivery systems becomes paramount. Qian Li et al. have shown that a peritoneal-localized hydrogel is based on an artificial exosome derived from engineered M1-type macrophages. This hydrogel can regulate peritoneal macrophages’ polarization and phagocytic function following X-ray radiation-induced immunogenicity, enhancing phagocytosis and antigen presentation to ovarian cancer cells [130].

Despite significant advancements, there are still considerable hurdles to surmount before exosome-based therapies can be effectively translated from bench to bedside. The intricate nature of exosome biology, challenges in large-scale production, and the need for precise targeting mechanisms are among the hurdles that need to be addressed to realize the full therapeutic potential of exosomes in the fight against ovarian cancer.

## 6. Conclusions

Exosomes emerge as central orchestrators of the molecular events that drive peritoneal metastasis and drug resistance in ovarian cancer. This discussion has highlighted the complex roles of exosomes in modulating key processes such as epithelial-to-mesenchymal transition (EMT), which primes cancer cells for metastasis, and the re-engineering of the extracellular matrix, which clears the path for tumor invasion. Furthermore, exosomes are implicated in fostering new blood vessel formation, ensuring the metastatic cells’ survival and spread. On the front of drug resistance, exosomes are revealed to be conveyors of chemoresistance, facilitating the horizontal transfer of drug-resistant genetic material and modulating cell survival pathways to evade the cytotoxic effects of therapy. This review has underscored the intricate molecular mechanisms at play, suggesting that disrupting exosome-mediated communication could be a promising strategy for stalling the progression of ovarian cancer. Further investigative efforts into these pathways may unlock new avenues for therapeutic intervention, potentially leading to more effective management of the disease and improved patient survival rates.

## Figures and Tables

**Figure 1 biomolecules-14-01099-f001:**
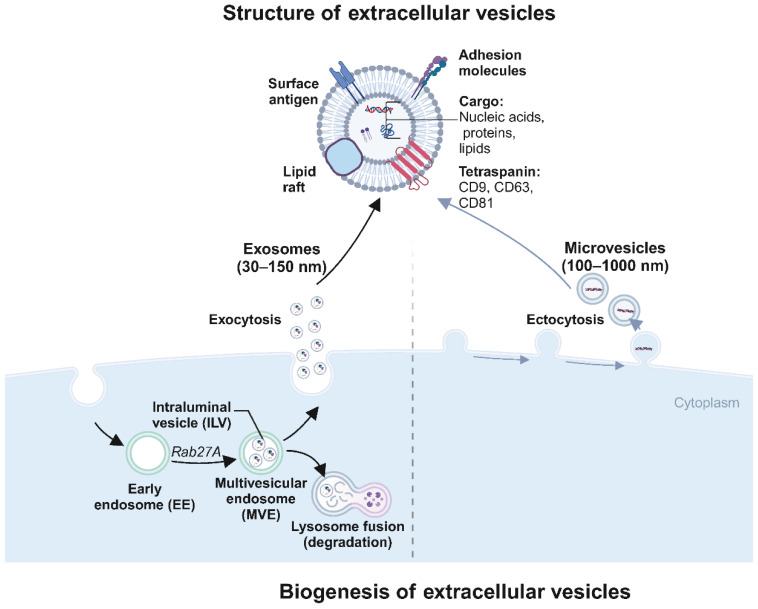
Biogenesis and structure of extracellular vesicles (EVs) (created with BioRender.com (accessed on4 August 2024)). This diagram illustrates the biogenesis and structural characteristics of extracellular vesicles (EVs), specifically exosomes and microvesicles. Structure of extracellular vesicles: The diagram highlights EVs’ surface and internal components. EVs are lipid bilayer-enclosed structures displaying surface antigens (e.g., CD9, CD63), adhesion molecules, and cargo, including nucleic acids (DNA, RNA), proteins, lipids, and metabolites. Exosomes (30–150 nm in diameter) are typically depicted with a detailed cargo composition that facilitates various cellular functions and signaling pathways. Exosome formation: The formation begins within the cell cytoplasm, where early endosomes (EE) containing intraluminal vesicles (ILVs) mature into multivesicular endosomes (MVE) under the influence of Rab27A. MVEs then fuse with lysosomes for degradation or fuse with the plasma membrane to release ILVs as exosomes into the extracellular space (exocytosis). Microvesicle formation: Microvesicles (100–1000 nm in diameter) are formed by the plasma membrane’s direct outward budding and fission (ectocytosis).

**Figure 2 biomolecules-14-01099-f002:**
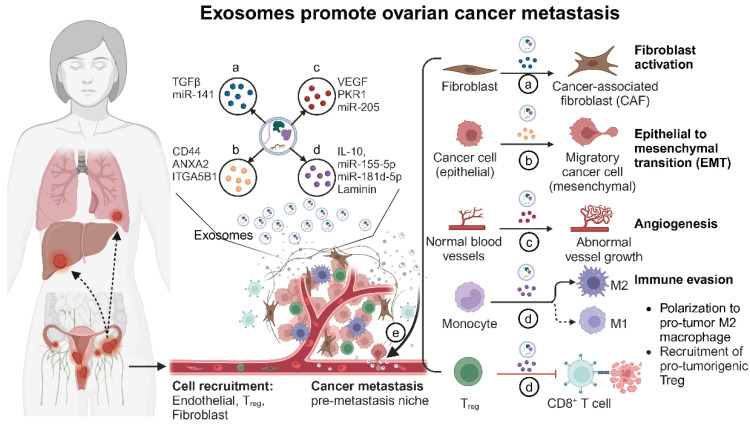
The role of exosomes in tumor microenvironment modulation and cancer metastasis (created with BioRender.com (accessed on 4 August 2024)). This figure illustrates the multifaceted roles of exosomes within the ovarian tumor microenvironment. Exosome functions: (**a**) Reprogramming fibroblasts into cancer-associated fibroblasts (CAFs): Ovarian cancer cell-derived exosomes carry TGFβ and other factors that transform normal fibroblasts into CAFs. CAFs then enhance tumor growth by remodeling the extracellular matrix and secreting growth factors. (**b**) Promoting epithelial–mesenchymal transition (EMT): Exosomes from ovarian cancer cells deliver signals such as TGFβ, miRNAs, and other EMT-promoting molecules to recipient epithelial cells. This induces EMT, characterized by the loss of epithelial markers and the gain of mesenchymal traits, which increases cell motility and invasiveness. (**c**) Facilitating angiogenesis: Exosomes from ovarian cancer cells promote angiogenesis by transferring pro-angiogenic factors such as VEGF, miRNAs, and other signaling molecules to endothelial cells. This results in the formation of new blood vessels, which supply the growing tumor with nutrients and oxygen. (**d**) Modulating immune response: Exosomes influence immune cells by promoting immunosuppressive T_REG and MDSCs while inhibiting cytotoxic T-cells and NK cells, creating an immune-invasive environment for the tumor. (**e**) Facilitating metastasis: Exosomes prepare distant organs for tumor cell colonization by modifying local cells and creating pre-metastatic niches, thereby aiding in the spread of cancer cells.

**Figure 3 biomolecules-14-01099-f003:**
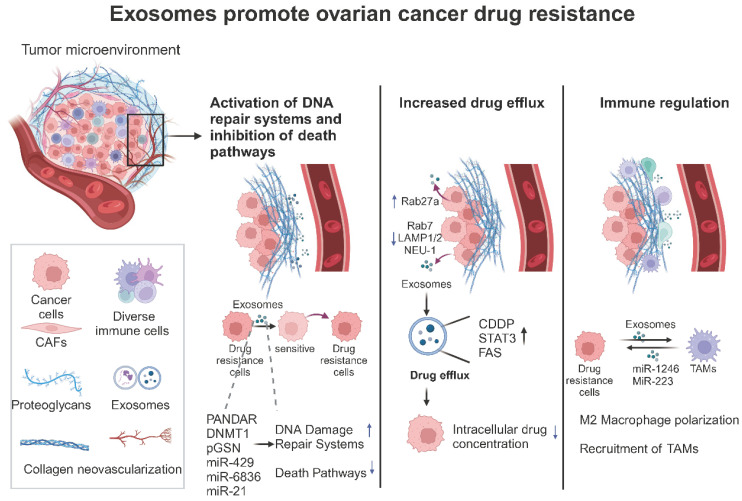
The role of exosomes in ovarian cancer drug resistance (created with BioRender.com (accessed on 22 August 2024)). This figure schematically depicts the three principal pathways by which exosomes in the tumor microenvironment promote drug resistance in ovarian cancer. These are as follows: (1) the activation of the mid-tumor DNA repair system and inhibition of the death pathways; (2) direct promotion of drug efflux to reduce intracellular drug concentrations in tumor cells; and (3) direct modulation of immune cells to create an immunosuppressive microenvironment.

**Table 1 biomolecules-14-01099-t001:** Exosomes promote ovarian cancer metastasis.

Exo-Source	Content	Target Cells	Mechanism	Effect	Refer.
Cancer cells	CD44	HPMCs	EMT and secrete MMP9	EMT	[37]
Cancer cells	ANXA2	HMRSV5	PI3K/AKT/mTOR pathway	EMT	[38]
Serum and Ascites	ITGA5B1/AEP	Mesothelial cells	FAK/Akt/Erk pathway	EMT	[39]
Cancer cells	PKR1	HUVECs	STAT3/PKR1 signaling pathway	Angiogenesis	[40]
Cancer cells	miR-205	HUVECs	PTEN-AKT pathway	Angiogenesis	[41]
Ascites	sE-cad	HUVECs	β-catenin and NF-κB signaling pathways	Angiogenesis	[42]
Cancer cells	miR-141	Stromal fibroblasts	Hippo signaling pathway	Reprogramming CAFs	[3]
CAFs	miR-29c-3p	Cancer cells	Secrete MMP2	Regulation of CAFs	[43]
CAFs	miR-21	Cancer cells	Downregulation of APAF1	Regulation of CAFs	[44]
Ascites	Biomolecule	T-cells	Inhibition of T-cell proliferation and activation	Immune suppression	[45]
Cancer cells	miR-155-5p	T-cells	PD-L1	Immune suppression	[46]
Cancer cells	miR-181d-5p	TAMs	M2-type macrophage polarization	Immune suppression	[47]
Cancer cells	Laminin	TAMs	ntegrinαvβ5/akt/sp1 pathway and M2-type macrophage polarization	Immune suppression	[48]
TAMs	miR-221-3p	Cancer cells	Inhibits CDKN1B	Immune suppression	[49]
TAMs	miR-29a-3p	Cancer cells	FOXO3-AKT/GSK3β/PD-L1 pathway	Immune escape	[50]
TAMs	miR-21-5p	T-cells	STAT3 inhibition and Treg/Th17 ratio imbalance	Immune suppression	[51]

**Table 2 biomolecules-14-01099-t002:** Exosomes promote drug resistance in ovarian cancer.

Exo-Source	Content	Target Cells	Mechanism	Effect	Refer.
Cancer cells	Cisplatin	N/A	Transmembrane protein TMEM205	Drug Efflux	[99,100]
Cancer cells	Cisplatin	N/A	Sorting and release	Drug Efflux	[101]
Cancer cells	Cisplatin, STAT3, and FAS	N/A	Rab27a, Rab7, LAMP1/2 and NEU-1	Drug Efflux	[102]
Cancer cells	DNMT1	Sensitive cells	DNA replication and damage repair	DNA repair	[103]
Cancer cells	miR-429	Sensitive cells	CASR/STAT3 pathway	DNA repair	[104]
Cancer cells	hsa_circ_0010467	Sensitive cells	LIF/STAT3 pathway	DNA repair	[105]
CAAs and CAFs	miR-21	Cancer cells	Down-regulating APAF1	Inhibiting apoptosis	[44]
CAFs	miR-98-5p	Cancer cells	Down-regulating CDKN1A	Inhibiting apoptosis	[106]
Cancer cells	miR-21-3p and miR-891-5p	Sensitive cells	DNA repair mechanisms	Inhibiting apoptosis	[107]
Cancer cells	miR-1246	TAMs	M2-type macrophage polarization	Immune suppression	[108]
TAMs	miR-223	Cancer cells	PTEN-PI3K/AKT pathway	Immune suppression	[109]

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
