# Peer review of "Exosomes: Key Factors in Ovarian Cancer Peritoneal Metastasis and Drug Resistance"

_biomolecules, 2024, doi:10.3390/biom14091099_

Round 1

Reviewer 1 Report

Comments and Suggestions for Authors

This manuscript is a comprehensive review article summarizing the current efforts on the involvement of exosome-mediated mechanisms in ovarian cancer metastasis and drug resistance. The authors provide wide-ranging references including review and research articles as well as controversial results for discussion, which is appreciated. It would be ideal if reducing the amount of review articles included in the reference though (appears more than 50 references are review articles). In general, the manuscript is well-written and easy to follow. Few suggestions for improving the reading flow are listed below.

1.       Suggest shortening the introduction, reducing the detailed descriptions overlapping with other sections and consolidating references. Lines 113-115 is one example of where descriptions overlap with Introduction with new references.

2.       Suggest moving Lines 131-149 to next section as they are more detailed and relevant to mechanism rather than biological characteristics.

3.       Line 160 - Figure 2 also describes mechanisms other than immune response. Suggest revising "immune response" to "tumor microenvironment modulation and cancer metastasis" to align with the Figure title.

4.       Suggest moving Section 3.3 to 3.1 to match with the order shown in Figure 2.

5.       Consider breaking metastasis and drug-resistance into two sections, meaning making Section 3.5-3.7 a new section 4. It would be nice to have a new Figure for drug-resistance mechanisms similar to Figure 2 for metastasis. If drug-resistance is going to be a new section, Table 2 and potential new figure can be grouped into this new section for a better reading flow.

Author Response

  1. Suggest shortening the introduction, reducing the detailed descriptions overlapping with other sections and consolidating references. Lines 113-115 is one example of where descriptions overlap with Introduction with new references.

    Response: Thank you for your valuable feedback and constructive suggestions. We've taken your insights into account and have made the introduction more concise. This was achieved by eliminating detailed descriptions that were redundant with content in other sections. Specifically, we've removed lines 50-53 and lines 62-64, as they were repetitive of lines 113-115.

    1. Suggest moving Lines 131-149 to next section as they are more detailed and relevant to mechanism rather than biological characteristics.

    Response: Thanks for your suggestion. We move lines 131-143 to the next section, see lines 160-173.

    1. Line 160 - Figure 2 also describes mechanisms other than immune response. Suggest revising "immune response" to "tumor microenvironment modulation and cancer metastasis" to align with the Figure title.

    Response: Thank you for highlighting the discrepancy. We value your meticulousness. Following your recommendation, we have updated the phrase on line 160 from "immune response" to "tumor microenvironment modulation and cancer metastasis," ensuring it accurately reflects the title of Figure 2.

    1. Suggest moving Section 3.3 to 3.1 to match with the order shown in Figure 2.

    Response: Thank you for your feedback on the manuscript structure. We have repositioned Section 3.3 to Section 3.1 for consistency with the sequence depicted in Figure 2. Your input has been instrumental in improving the coherence of our work.

    1. Consider breaking metastasis and drug-resistance into two sections, meaning making Section 3.5-3.7 a new section 4. It would be nice to have a new Figure for drug-resistance mechanisms similar to Figure 2 for metastasis. If drug-resistance is going to be a new section, Table 2 and potential new figure can be grouped into this new section for a better reading flow.

    Response: Your guidance on the structure of the manuscript has been invaluable, particularly concerning the sections on metastasis and drug resistance. Following your advice, we've reorganized Sections 3.5 to 3.7 into a newly created Section 4, focusing exclusively on drug resistance (see lines 470-493).

    Additionally, we've introduced a new figure (Figure 3) that delineates the drug resistance mechanisms, complementing the earlier figure (Figure 2) that detailed metastasis mechanisms. To enhance readability, we've also integrated Table 2 and Figure 3 into this new section.We are confident that these modifications will enhance the clarity and structure of our manuscript. We would like to express our gratitude once more for your perceptive feedback.

Reviewer 2 Report

Comments and Suggestions for Authors

The paper by Shao et al. al is a review describing the different molecular mechanisms by which exosomes contribute to peritoneal metastasis and drug resistance in ovarian cancer.  

The information summarized in this review is very interesting and can be useful for many researchers. The manuscript is well written and clear. The quality of the figures is very good, and tables keep to publication standards of the journal. References extensively cover all aspects of the review. Citations are correct and follow a consistent format.

In my opinion the manuscript can be published in its present form with few minor corrections.

I would suggest including epithelial-mesenchymal transition as a keyword.

It is recommended avoiding abbreviations in the abstract (EMT)

Author Response

I would suggest including epithelial-mesenchymal transition as a keyword.

It is recommended avoiding abbreviations in the abstract (EMT)

Response: We appreciate your insightful recommendations and have acted accordingly. The term "epithelial-mesenchymal transition" has now been added. Furthermore, we have carefully revised the abstract to eliminate the use of the abbreviation "EMT." This change is intended to enhance the abstract's clarity.

Reviewer 3 Report

Comments and Suggestions for Authors

The manuscript entitled “Exosomes: Key Factors in Ovarian Cancer Peritoneal Metastasis 2 and Drug Resistance” by Shao et. al, provided a comprehensive review of the role that exosomes play during the progression of ovarian cancer, especially metastasis and drug resistance. The manuscript is well-structured, including introducing mechanisms on exosome generation and their roles in EMT and expelling the chemo drugs during chemotherapy. This review is of broad interest to the readership of Biomolecules, I would recommend the review article be published after the minor issue below is addressed:

1.      It would be better also to include some potential therapeutic strategies to tackle the problem brought by exosomes during the treatment of ovarian cancer. If certain information is lacking, it would also be beneficial to include authors’ comments on further detailed direction to solve these problems.

Author Response

  1. It would be better also to include some potential therapeutic strategies to tackle the problem brought by exosomes during the treatment of ovarian cancer. If certain information is lacking, it would also be beneficial to include authors’ comments on further detailed direction to solve these problems.

Response: We are deeply appreciative of your thoughtful feedback and the constructive critique you've provided on our manuscript. Pursuant to your suggestions, we have added a new section titled "The Potential of Exosomes in the Treatment of Ovarian Cancer" (lines 621-657). This section delves into the diverse therapeutic strategies that could be leveraged to overcome the challenges exosomes present in the treatment of ovarian cancer. It comprehensively discusses the prospective targeting of exosomes and explores innovative methods, including the utilization of exosomes for drug delivery, immune system modulation, and the latest technologies being developed to combat peritoneal metastases. We are confident that this new section not only enriches the manuscript but also opens up exciting possibilities for future research directions and clinical interventions. We extend our sincere thanks for your invaluable input.

Reviewer 4 Report

Comments and Suggestions for Authors

1-  This is a well written review addressing addressing Exosomes as key factors in the import ovarian cancer metastasis and drug resistance.

2-  Line-  Line 144 "illuminated"   "Emphasized" may be a better description.

3-  Figures 1 and 2, are clear and well constructed, and emphasize the role of Vesicles and Exosomes.

4-  Line 305, Tumor-associated fibroblasts is abbreviated as (CAFs)? This is evident on subsequent lines.  I am not sure about he reason for this abbreviation? I am assuming that" Tumor is cancer"; however in English, tumor may not be cancer unless it metastasizes, and therefore could be misleading. Why not abbreviate it "TAF"?.

5-  Otherwise, it is an excellent review. 

Author Response

  1. Line 144 "illuminated"   "Emphasized" may be a better description.

Response: Thank you for your input on the wording in line 144. We concur that "emphasized" is more appropriate than "illuminated" and have updated the manuscript accordingly. Your attention to detail is greatly valued.

  1.  Line 305, Tumor-associated fibroblasts is abbreviated as (CAFs)? This is evident on subsequent lines.  I am not sure about he reason for this abbreviation? I am assuming that" Tumor is cancer"; however in English, tumor may not be cancer unless it metastasizes, and therefore could be misleading. Why not abbreviate it "TAF"?.

Response: Thank you for your guidance on our manuscript's terminology. We've replaced "tumor-associated fibroblasts" with "cancer-associated fibroblasts" (CAFs) throughout, aligning the terminology with standard usage and enhancing clarity. Your input is highly appreciated.